# Wnt10b-GSK3β–dependent Wnt/STOP signaling prevents aneuploidy in human somatic cells

Yu-Chih Lin[1], Alexander Haas[1], Anja Bufe[2] ®, Sabnam Parbin[3], Magdalena Hennecke[1], Oksana Voloshanenko[4], Julia Gross[3], Michael Boutros[4] ®, Sergio P Acebron[2], Holger Bastians[1] ®

**Wnt signaling is crucial for proper development, tissue homeostasis and cell cycle regulation. A key role of Wnt signaling is the GSK3β-mediated stabilization of β-catenin, which mediates many of the critical roles of Wnt signaling. In addition, it was recently revealed that Wnt signaling can also act independently of β-catenin. In fact, Wnt mediated stabilization of proteins (Wnt/STOP) that involves an LRP6-DVL–dependent signaling cascade is required for proper regulation of mitosis and for faithful chromosome segregation in human somatic cells. We show that inhibition of Wnt/LRP6 signaling causes whole chromosome missegregation and aneuploidy by triggering abnormally increased microtubule growth rates in mitotic spindles, and this is mediated by increased GSK3β activity. We demonstrate that proper mitosis and maintenance of numerical chromosome stability requires continuous basal autocrine Wnt signaling that involves secretion of Wnts. Importantly, we identified Wnt10b as a Wnt ligand required for the maintenance of normal mitotic microtubule dynamics and for proper chromosome segregation. Thus, a self-maintaining Wnt10b-GSK3β–driven cellular machinery ensures the proper execution of mitosis and karyotype stability in human somatic cells.**

## Introduction

Wnt signaling pathways play critical roles in various developmental processes during embryogenesis and are important for the maintenance of adult tissue homeostasis (1). Wnt signaling has also a major role in cell proliferation, cell survival and differentiation and its misregulation is closely linked to human diseases including developmental pathologies and cancer (2). A variety of different Wnt ligands are expressed in human cells and secreted into the extracellular space to activate Wnt signaling in an auto- or paracrine manner (3). Secretion of Wnts requires their palmitoylation by the O-acyltransferase Porcupine and transport by the general Wnt cargo receptor Evi/Wntless (Wls) (4, 5, 6). Upon secretion, different Wnts bind to different Frizzled (FZD) receptors and activate intracellular signaling, which can be categorized into canonical/β-catenin–dependent and non-canonical/β-catenin–independent pathways (2).

The best characterized Wnt signaling pathway is the so-called canonical pathway that leads to the activation of the transcription factor β-catenin. In this pathway, Wnt ligands such as Wnt3a bind to different Frizzled (FZD) receptors and the co-receptors LRP5/6 (7, 8). The ligand receptor complex clusters together with Dishevelled (Dvl) proteins in LRP6 signalosomes that recruit and inactivate a cytoplasmic destruction complex consisting of adenomatous polyposis coli, AXIN1 and the kinases CK1α and GSK3β (8, 9, 10). The inactivation of this complex prevents the ongoing phosphorylation of the transcription factor β-catenin by CK1α and GSK3β otherwise leading to its degradation by the ubiquitin–proteasome pathway (10, 11, 12). Hence, activation of the classical canonical Wnt signaling results in stabilization of β-catenin, which subsequently activates the expression of various target genes that are involved in the different cellular and developmental outcomes. It is known that different secreted Wnt ligands can function as canonical Wnts stabilizing β-catenin, albeit at different strength (3). As a counterpart, the secreted protein Dickkopf-1 (DKK1) inhibits canonical Wnt signaling by binding to and inducing the turnover of LRP5/6 co-receptors (13).

Interestingly, canonical Wnt signaling was shown to be involved in cell cycle regulation. A crucial target of β-catenin is c-myc, which drives the expression of *CNND1* (encoding for cyclin D1) as a key driver of the G1/S transition of the cell cycle (14, 15). Moreover, in mouse embryonic stem cells (mESCs), the loss of Evi/Wls, which is associated with a general lack of Wnt secretion (16), was found to impact mitotic progression and resulted in genome instability (17).

More recently, it became clear that Wnt/LRP6 signaling can also stabilize proteins other than β-catenin in a GSK3β-dependent

[1]Georg-August University Göttingen, Göttingen Center for Molecular Biosciences (GZMB) and University Medical Center Göttingen (UMG), Institute of Molecular Oncology, Section for Cellular Oncology, Göttingen, Germany   [2]University of Heidelberg, Centre for Organismal Studies (COS), Heidelberg, Germany   [3]University Medical Center Göttingen (UMG), Hematology and Oncology and Developmental Biochemistry, Göttingen, Germany   [4]Department of Cell and Molecular Biology, German Cancer Research Center (DKFZ), Division of Signaling and Functional Genomics and Heidelberg University, Medical Faculty Mannheim, Heidelberg, Germany

Correspondence: holger.bastians@uni-goettingen.de
Julia Gross's present address is HMU Health and Medical University Potsdam, Potsdam, Germany

manner by sequestering the destruction complex in multivesicular bodies (10). This Wnt signaling pathway, now referred to as Wnt-mediated stabilization of proteins (Wnt/STOP) also involves LRP co-receptors and DVL (10, 18, 19, 20, 21) and promotes cell division and growth (18, 22, 23, 24). Wnt/STOP activity peaks at the G2/M transition of the cell cycle because of the kinases CDK14-16, which phosphorylate and activate LRP6, and CDK1, which, in turn, phosphorylates and recruits BCL9 to the mitotic LRP6 signalosome (22, 25).

Our previous work has demonstrated that loss of LRP5/6 or DVL, but not of $\beta$-catenin deregulates mitosis and causes chromosome missegregation suggesting that Wnt/STOP is required for faithful mitosis. In fact, we showed that loss of LRP6/DVL causes abnormally increased microtubule growth rates in mitotic spindles that act as a trigger for the generation of erroneous microtubule-kinetochore attachments leading to whole chromosome missegregation (26, 27). However, whether proper mitosis requires autocrine Wnt signaling and whether specific Wnt ligands are involved has not been addressed so far. In our work presented here, we show that basal canonical Wnt signaling involving LRP6 and leading to continuous suppression of GSK3$\beta$ activity is required for the proper regulation of mitotic microtubule dynamics and chromosome segregation. We find that autocrine Wnt signaling requiring palmytoylation and secretion of Wnt ligands is essential for the proper execution of mitosis. Surprisingly, we identified Wnt10b as crucial Wnt ligand involved in the regulation of mitosis and required for preventing whole chromosome missegregation.

# Results

### Inhibition of Wnt signaling at the receptor level causes abnormal microtubule dynamics and aneuploidy in human somatic cells

To inhibit canonical Wnt signaling at the receptor level we treated chromosomally stable human HCT116 cells with DKK1, an inhibitor of the Wnt co-receptor LRP6 (13, 28). HCT116 cells harbor a heterozygous stabilizing mutation in $\beta$-catenin, but these cells still respond to Wnt in reporter assays (16, 29). Accordingly, we found that Wnt3a further increased and DKK1 treatment reduced basal TCF4/$\beta$-catenin reporter activity. Co-treatment with the GSK3$\beta$ inhibitor CHIR99021 reverted this effect (Fig S1A). Upon inspection of DKK1-treated cells progressing through mitosis by live cell microscopy, we found evidence for chromosome missegregation as indicated by the appearance of lagging chromosomes during anaphase and by generation of micronuclei upon exit from mitosis (Fig S1B). Rigorous quantification of the proportion of cells exhibiting lagging chromosomes indeed revealed a significant induction of lagging chromosomes upon 16-h DKK1 treatment, which was suppressed by co-treatment with a low concentration of CHIR99021 (Fig 1A), which did not affect cell viability (Fig S1C). Our previous work has linked the generation of lagging chromosomes to abnormally increased microtubule assembly rates in mitotic spindles in response to Wnt inhibition (26). Hence, we used live cell microscopy to track GFP-tagged microtubule end binding protein 3 (EB3) at microtubule plus tips in DKK1 treated cells synchronized in prometaphase of mitosis

(Fig S1D and Video 1). DKK1 treatment significantly increased mitotic microtubule growth rates in HCT116 cells (Fig 1B). In line with this, non-transformed human retina pigment epithelial cells (RPE1-hTert) also exhibited increased microtubule assembly rates upon DKK1 treatment (Fig S2A). As demonstrated previously (26, 30), increased microtubule growth rates can be efficiently suppressed by treatment with sub-nanomolar concentrations of Taxol (Fig 1B), a microtubule binding drug known to inhibit microtubule dynamics (31). Upon DKK1 treatment restoration of proper microtubule growth rates by low-dose Taxol prevented the generation of lagging chromosomes (Fig 1A), suggesting a direct link between increased microtubule growth rates and the induction of whole chromosome missegregation in human somatic cells after DKK1-mediated LRP6 inhibition.

Because perpetual chromosome missegregation leads to an evolvement of aneuploidy, we investigated whether continuous inhibition of canonical Wnt signaling causes a high chromosome number variability in a microtubule dynamics-dependent manner. For this, we generated single-cell clones grown for 30 generations in the presence or absence of inhibitors as illustrated in Fig 1C and determined the proportion of cells harboring chromosome numbers deviating from the modal (45 chromosomes in HCT116 cells and 46 chromosomes in RPE1 cells). Continuous DKK1 treatment resulted in an evolvement of aneuploidy, which was suppressed by concomitant inhibition of GSK3$\beta$, both, in HCT116 (Fig 1D and E) and in RPE1 cells (Fig S2B and C). Co-treatment with low doses of Taxol, which restores proper microtubule growth rates, efficiently suppressed the generation of aneuploidy over time, indicating that abnormally increased microtubule dynamics is responsible for perpetual chromosome missegregation after Wnt inhibition (Fig 1D and E). Our results strongly corroborate the finding that inhibition of canonical Wnt signaling, which comprises LRP6 and increased GSK3$\beta$ activity results in abnormally increased microtubule growth rates during mitosis leading to whole chromosome missegregation and aneuploidy. Thus, human somatic cells require basal Wnt signaling to maintain proper microtubule dynamics in mitosis, faithful chromosome segregation, and a stable karyotype.

### Abnormal microtubule dynamics and chromosome missegregation is triggered by increased GSK3$\beta$ activity

The GSK3$\beta$ kinase is the critical downstream component of the canonical Wnt-regulated destruction complex (1). Because abnormally increased microtubule growth rates and chromosome missegregation upon DKK1 treatment were suppressed by GSK3$\beta$ inhibition (Fig 1), we asked whether increased GSK3$\beta$ activity is sufficient to trigger the mitotic errors. To this end, we generated two independent stable cell lines derived from HCT116 cells showing low-to-moderate overexpression of GSK3$\beta$ (Fig 2A). These cell lines were subjected to EB3 tracking experiments to determine microtubule growth rates in living mitotic cells. Clearly, increasing GSK3$\beta$ levels were sufficient to increase microtubule assembly rates to the same levels seen upon DKK1 treatment. The abnormal microtubule growth rates were restored to control levels by treatment with low-doses of Taxol and by small molecule–mediated GSK3$\beta$ inhibition (CHIR99021) indicating that the kinase activity is responsible for the observed phenotypes (Fig 2B). Accordingly, increased GSK3$\beta$ levels

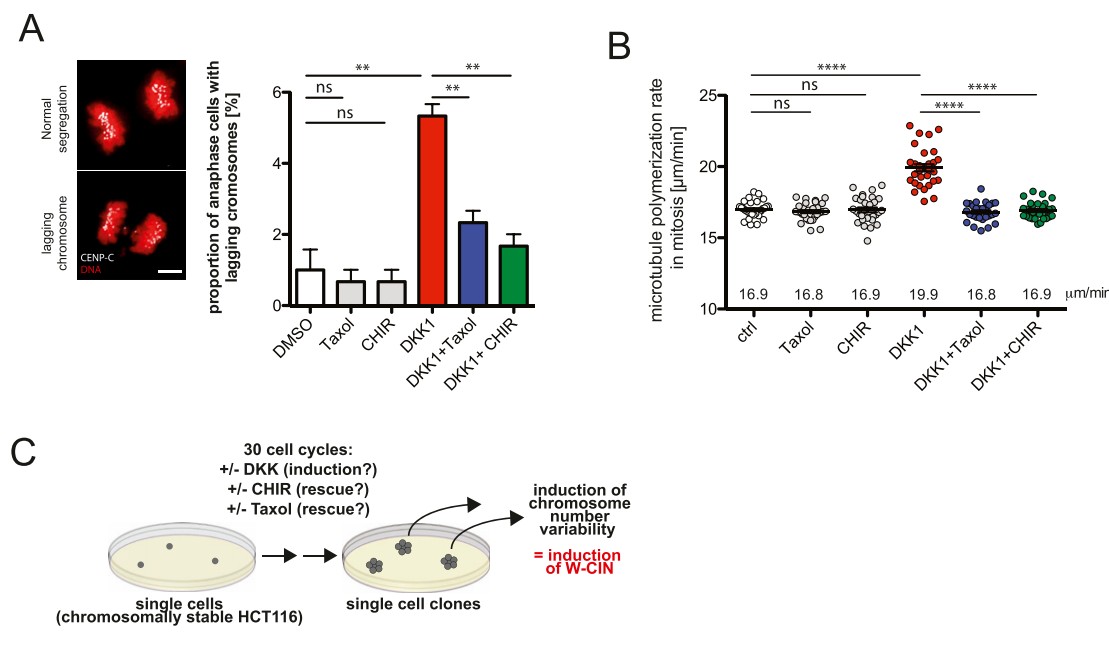

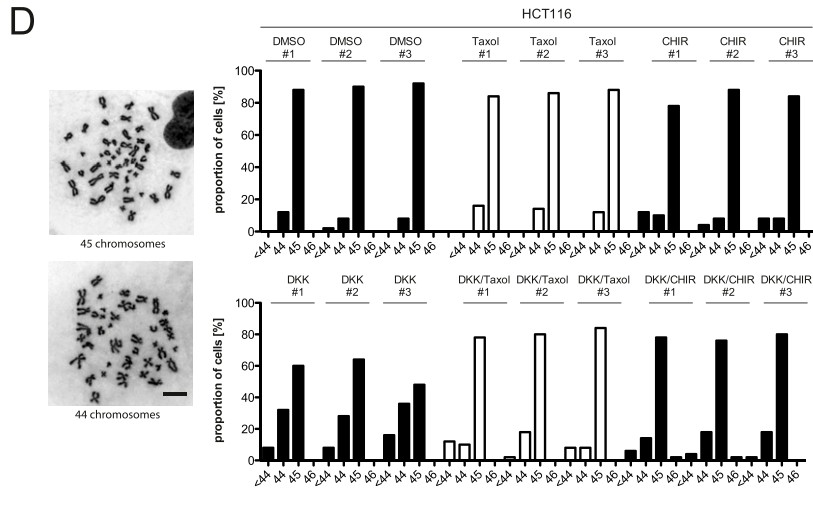

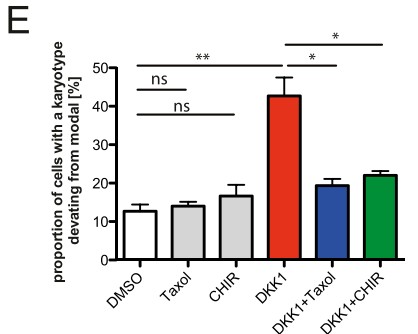

**Figure 1.  Inhibition of Wnt signaling at the receptor level causes abnormal microtubule dynamics and aneuploidy in human somatic cells.**
**(A)** Detection and quantification of anaphase cells with lagging chromosomes. HCT116 cells were treated with 600 ng/ml DKK1, 0.2 nM Taxol or 0.6 $\mu$M CHIR99021 (CHIR) for 16 h and lagging chromosomes were detected by immunofluorescence microscopy. Representative images of cells with or without lagging chromosomes are shown (chromosomes, Hoechst 33342, red; kinetochores, Cenp-C, white; scale bar, 5 $\mu$M). Only kinetochore-positive chromosomes were counted as lagging chromosomes. The graph shows the quantification of the proportion of cells exhibiting lagging chromosomes (mean ± SEM, *t* test, n = 300 cells from three independent experiments). **(A, B)** Mitotic microtubule plus-end assembly rates in HCT116 cells treated as in (A). Scatter dot plots show average assembly rates (20 microtubules/cell, mean ± SEM, *t* test,

are sufficient to induce lagging chromosomes during anaphase, which were also suppressed by Taxol or by GSK3$\beta$ inhibitor treatment (Fig 2C). To test whether an increase in GSK3$\beta$ can trigger the development of aneuploidy, we generated single-cell clones from control and GSK3$\beta$ overexpressing cells and determined chromosome number variability that evolved within 30 d of cultivation. In fact, overexpression of GSK3$\beta$ in otherwise chromosomally stable cells was found to be sufficient to trigger the evolvement of aneuploidy (Fig 2D). Thus, increased GSK3$\beta$ activity upon inhibition of canonical Wnt signaling represents a relevant trigger for abnormal mitotic microtubule dynamics, chromosome missegregation, and aneuploidy. Because GSK3$\beta$ represents the kinase in the Wnt signaling pathway that phosphorylates and destabilizes not only $\beta$-catenin, but also additional proteins (10), it seems likely that Wnt/STOP is required for the normal regulation of microtubule dynamics.

## Wnt secretion ensures normal microtubule dynamics during mitosis and prevents aneuploidy

Next, we focused on the upstream events of Wnt signaling involved in mitotic regulation and asked whether continuous autocrine Wnt signaling is required to suppress aneuploidy in human somatic cells. To address this question, we generated Evi/Wls-knockout cells derived from chromosomally stable HCT116 cells using CRISPR/cas9 (HCT116$^{Evi\_KO}$). HCT116 cells were transfected with gRNA targeting Evi/Wls along with a Cas9 expression plasmid as described (32). Three independent knockout cell clones were isolated. Further details on the characterization of the HCT116$^{Evi\_KO}$ cells will be provided elsewhere (Voloshanenko et al, manuscript in preparation). Evi/Wls deficiency in the three knockout cell clones was verified by Western blotting (Fig 3A). The HCT116$^{Evi\_KO}$ cells were used to determine microtubule growth rates in mitosis by live cell analyses. All three cell clones exhibited increased microtubule assembly rates as seen for DKK1 treated cells, which were restored to normal levels by low-dose Taxol treatment and also upon GSK3$\beta$ inhibition indicating that abnormal microtubule dynamics induced by Evi/Wls knockout is due to global loss of Wnt signaling (Fig 3B). Consequently, loss of Evi/Wls causes a significant increase in cells with lagging chromosomes and this defect was also suppressed by Taxol or GSK3$\beta$ inhibitor treatment (Fig 3C). In line with the abnormal microtubule dynamics and lagging chromosome induction, karyotype analyses revealed that single cell clones derived from the three independent HCT116$^{Evi\_KO}$ cell lines developed aneuploidy within 30 generations (Fig 3D). Thus, loss of Wnt secretion causes the same mitotic defects leading to aneuploidy as seen upon LRP6 inhibition. To further support this finding, we inhibited Porcupine, which is the O-acyltransferase required for palmitoylation of Wnts and, thus, for their secretion (33). Porcupine inhibition by the small molecule inhibitor IWP12 (34) increased mitotic microtubule assembly rates (Fig 3E) and induced lagging chromosomes during

anaphase (Fig 3F) as seen in HCT116$^{Evi\_KO}$ cells. These data indicate that modification and secretion of Wnts is a prerequisite for normal mitotic microtubule dynamics, proper mitosis, and faithful chromosome segregation in human somatic cells.

## Identification of Wnt ligands crucial for proper execution of mitosis

To test whether proper mitotic regulation can be restored in HCT116$^{Evi\_KO}$ cells by paracrine Wnt signaling, the knockout cells were co-cultured with parental HCT116 cells, and microtubule growth rates were determined specifically in (EB3-GFP-transfected) HCT116$^{Evi\_KO}$ cells. In fact, abnormally increased microtubule assembly rates in Evi/Wls-deficient cells were corrected by co-culture with parental cells in a cell number–dependent manner indicating restoration of mitosis-relevant Wnt signaling by Wnt ligands derived from parental HCT116 cells (Fig 4A). Based on this, we asked whether specific Wnt ligands are able to suppress abnormally increased microtubule polymerization rates in HCT116$^{Evi\_KO}$ cells. For this, we generated conditioned media for 14 different Wnts, treated Evi/Wls-deficient cells with control or Wnt-conditioned media and determined microtubule assembly rates in living mitotic cells. This screening approach revealed that conditioned media for Wnt1, Wnt2, Wnt3, Wnt3a, Wnt5a, Wnt6, Wnt7a, Wnt8a, Wnt9b, Wnt10a, and Wnt16 had no effect on mitotic microtubule regulation (Fig 4B). However, addition of Wnt10b-conditioned medium fully restored proper mitotic microtubule polymerization rates in HCT116$^{Evi\_KO}$ cells, whereas addition of Wnt7b and Wnt9a-conditioned media resulted in partial rescues (Fig 4B). Thus, particularly Wnt10b, but possibly also Wnt7b and Wnt9a, represents candidate Wnt ligands involved in the regulation of mitotic microtubule dynamics and chromosome segregation.

## Basal Wnt10b signaling is required for proper mitotic regulation

Next, we wished to test whether basal endogenous levels of Wnt7a, Wnt9b, or Wnt10b are required for normal mitotic microtubule dynamics in HCT116 cells. We down-regulated a subset of endogenous Wnts, including Wnt7b, Wnt9a, and Wnt10b, which showed some rescue activity HCT116$^{Evi\_KO}$ cells, and determined mitotic microtubule dynamics. These five Wnts show highest expression levels in HCT116 cells (35), and siRNA-mediated down-regulation was achieved at similar levels for all of them (Fig S3A). Significantly, we found that only down-regulation of endogenous Wnt10b, but not of Wnt3a, 7b, 9a, and 16 was sufficient to increase microtubule polymerization rates, suggesting that Wnt10b is indeed the most relevant Wnt ligand involved in the regulation of mitosis in HCT116 cells (Fig 5A). Thus, although conditioned media containing high levels of Wnt7a and Wnt9b showed some rescue effect in HCT116$^{Evi\_KO}$ cells, basal levels of these Wnts seem not to be required for normal regulation of microtubule dynamics in HCT116

n = 30 cells). **(C)** Scheme illustrating the generation of single cell clones that were grown for 30 generations to analyze chromosome number variability in single cell clones. **(D)** Chromosome numbers per cell and aneuploidy in single-cell clones after the indicated treatments. For each single-cell clone, 50 metaphase spreads were analyzed and representative images for euploid and aneuploid karyotypes are given (scale bar, 10 $\mu$m). **(E)** Summary of aneuploidy induction in HCT116 cells after the indicated treatments. **(D)** The graph shows mean values ±SEM for three independent single cell clones depicted in (D).

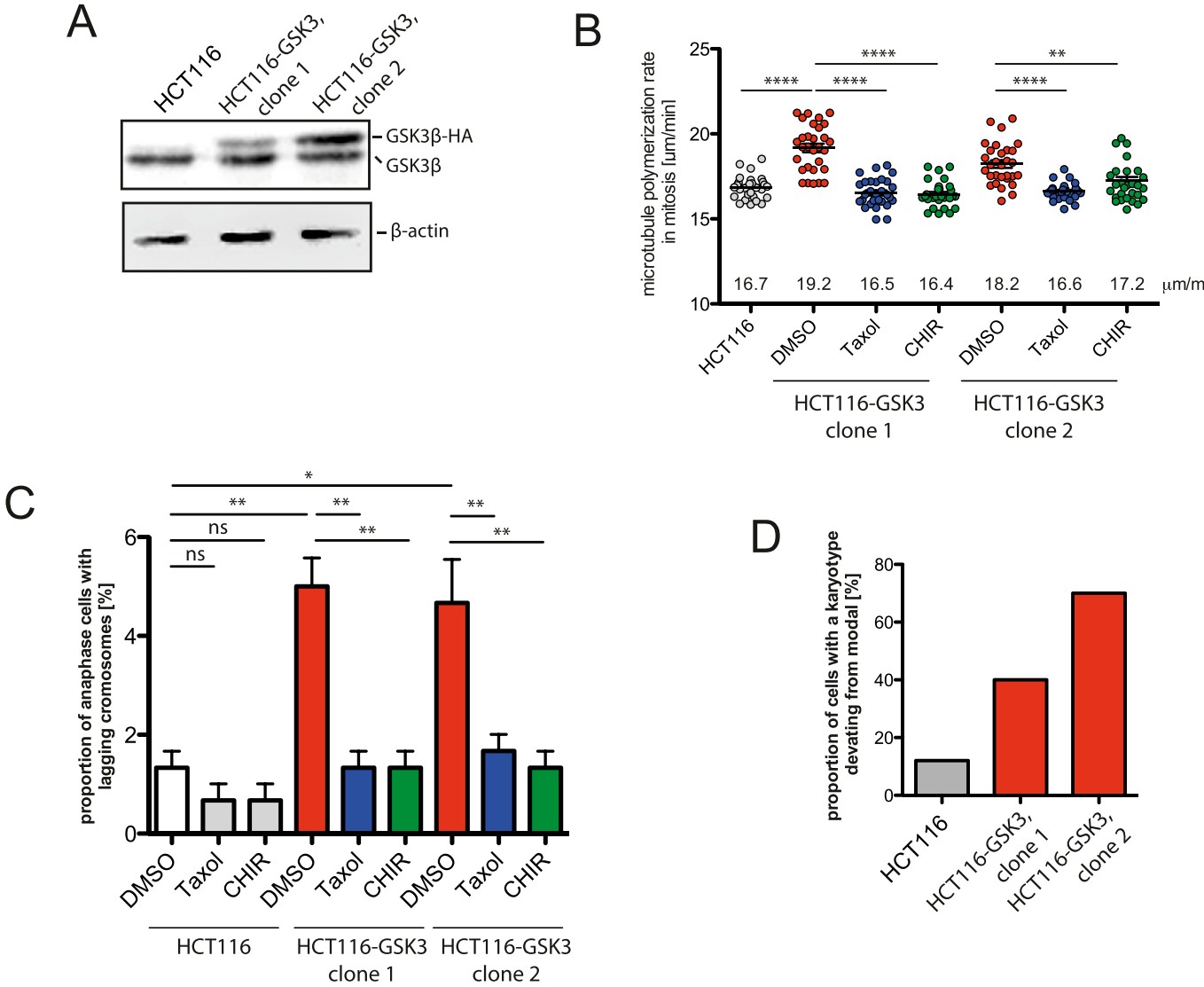

**Figure 2. Abnormal microtubule dynamics and chromosome missegregation is triggered by increased GSK3$\beta$ activity.**
**(A)** Stable GSK3$\beta$-HA overexpression in HCT116 cells. A representative Western blot shows endogenous and overexpressed HA-tagged GSK3$\beta$. $\beta$-actin served as a loading control. **(B)** Mitotic microtubule plus end assembly rates using two cell lines derived from HCT116 cells stably expressing GSK3$\beta$-HA and treated with DMSO (control), 0.2 nM Taxol, or 0.6 $\mu$M CHIR99021 for 16 h (20 microtubules/cell, mean ± SEM, t test, n = 30 cells). **(C)** Proportion of GSK3$\beta$-expressing HCT116 cells exhibiting lagging chromosomes. The graph shows mean values ±SEM (t test, n = 300 cells from three independent experiments). **(D)** Aneuploidy induction in HCT116 cells after overexpression of GSK3$\beta$. Two independent cell clones were analyzed, and chromosome numbers from 50 metaphase spreads were determined and the proportion of cells with chromosome numbers deviating from the modal was calculated.

cells. In addition to HCT116 cells, down-regulation of Wnt10b was also sufficient to increase mitotic microtubule polymerization rates in non-transformed RPE1-hTert cells (Fig S3B). After Wnt10b down-regulation, both the increase in mitotic microtubule polymerization as well as the resulting induction of lagging chromosomes were not only rescued by low-dose Taxol treatment, as expected (see e.g., Fig 1A and B), but also by addition of purified recombinant Wnt10b protein and also by partial inhibition of GSK3$\beta$ (Fig 5A and B) indicating specificity for Wnt10b and dependence on GSK3$\beta$ activity. Moreover, addition of purified recombinant Wnt10b, but not of Wnt3a fully rescued the abnormally increased mitotic microtubule polymerization rates and also chromosome missegregation in

HCT116[Evi_KO] cells (Fig 5C and D). Thus, Wnt10b, but not other canonical Wnts including Wnt3a, which activates $\beta$-catenin–dependent transcription stronger than Wnt10b in reporter assays in HCT116 cells (Fig S3C), is crucial for the regulation of mitotic chromosome segregation.

## Discussion

Our work indicates that Wnt signaling is pivotal for the proper progression mitosis and for the maintenance of genome stability. In particular, whole chromosome missegregation in mitosis resulting

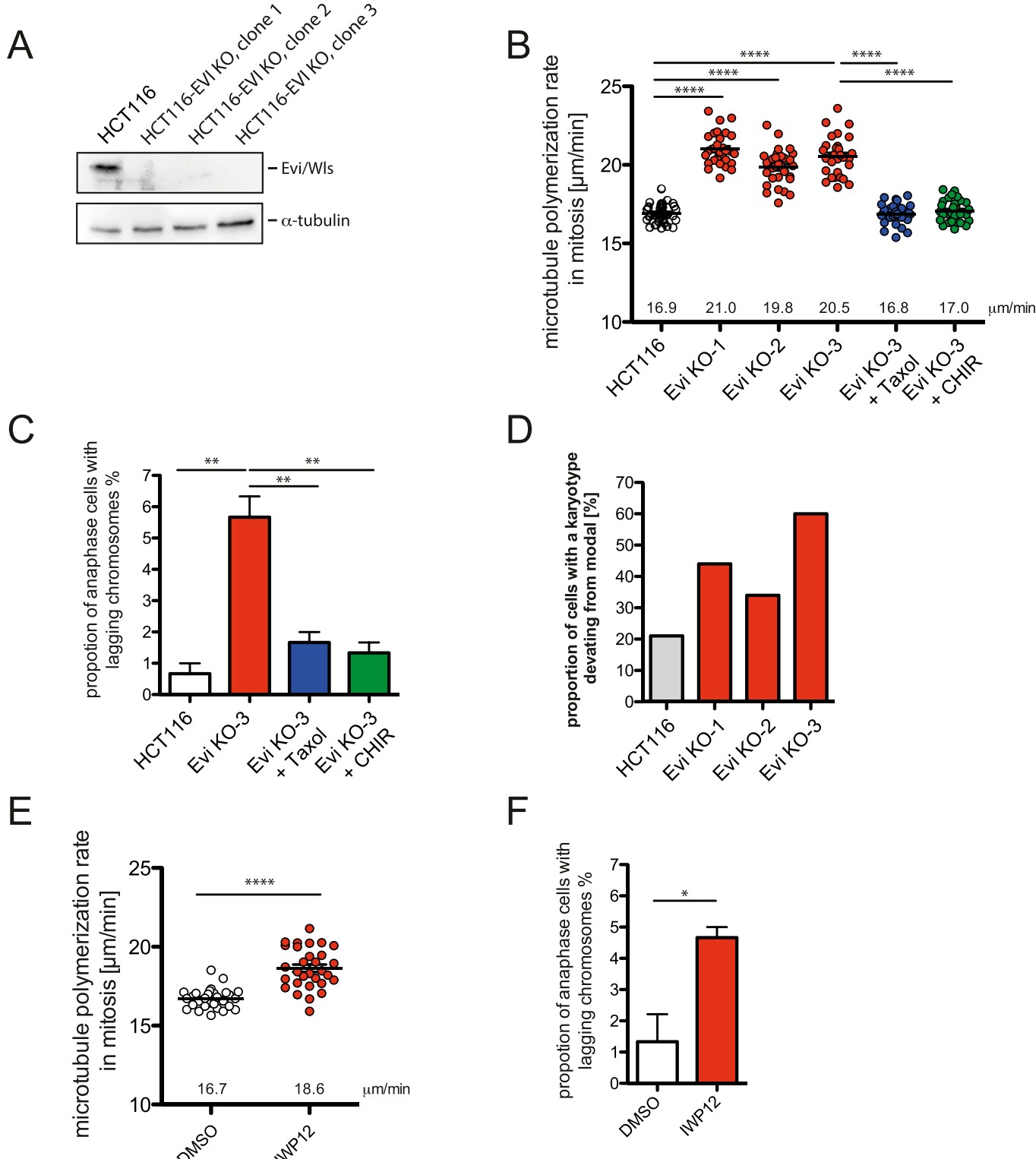

**Figure 3. Wnt secretion is required for normal microtubule dynamics during mitosis and to prevent aneuploidy.**
**(A)** Evi/Wls is not expressed in HCT116[Evi_KO] cells. A representative Western blot detecting Evi/Wls in parental and three HCT116[Evi_KO] cell lines is shown. **(B)** Mitotic microtubule plus end assembly rates in in parental and three HCT116[Evi_KO] cell lines and after treatment with DMSO (control), 0.2 nM Taxol, or 0.6 $\mu$M CHIR99021 for 16 h. Scatter dot plots show average assembly rates (20 microtubules/cell, mean ± SEM, $t$ test, n = 30 cells from three independent experiments). **(C)** Proportion of HCT116[Evi_KO] cells exhibiting lagging chromosomes. The graph shows mean values ±SEM ($t$ test, n = 300 cells from three independent experiments). **(D)** Induction of aneuploidy in HCT116[Evi_KO] single cell clones that were grown for 30 generations. Chromosome numbers from 50 metaphase spreads were determined, and the proportion of cells with chromosome numbers deviating from the modal was calculated. **(E)** Mitotic microtubule plus end assembly rates in HCT116 cells treated with 5 $\mu$M Porcupine inhibitor

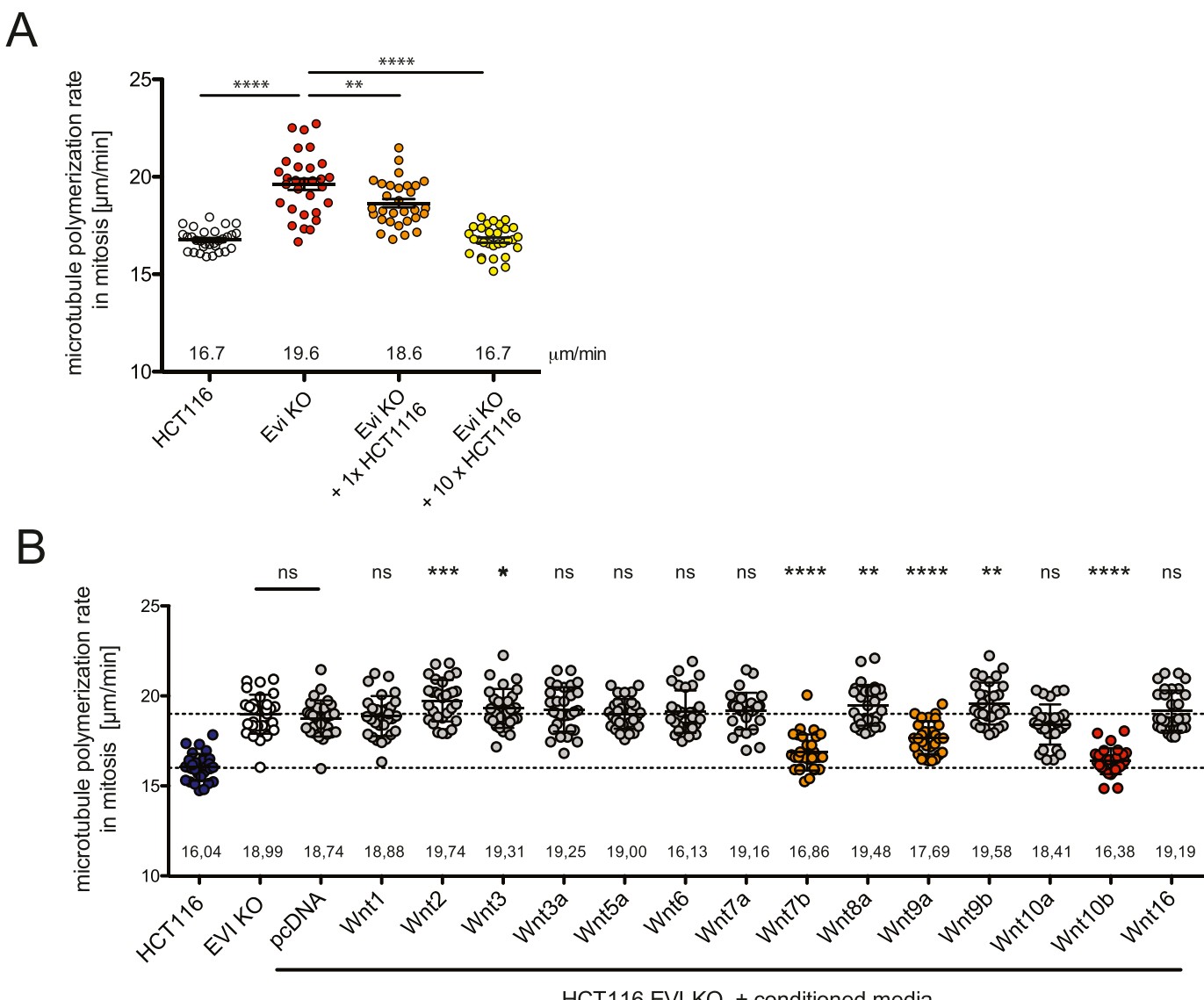

**Figure 4. Identification of Wnt ligands involved in the regulation of mitotic microtubule polymerization.**
**(A)** Mitotic microtubule plus end assembly rates in HCT116[Evi_KO] cells after co-culture with parental HCT116 cells. HCT116[Evi_KO] cells were transfected to express EB3-GFP and co-cultured with equal amount or 10-fold higher numbers of untransfected HCT116 cells for 48 h. HCT116[Evi_KO] cells were identified by EB3-GFP expression, and microtubule polymerization rates were measured in mitotic cells. Scatter dot plots show average microtubule growth rates (20 microtubules/cell, mean ± SEM, $t$ test, n = 30 cells from three independent experiments). **(B)** Mitotic microtubule plus end assembly rates in HCT116[Evi_KO] cells after treatment with different Wnt conditioned media. HCT116[Evi_KO] cells were transfected to express EB3-GFP and treated with conditioned media derived from transfected HEK293T cells as indicated for 16 h. Microtubule polymerization rates were measured in mitotic cells. Scatter dot plots show average microtubule growth rates (20 microtubules/cell, mean ± SEM, $t$ test, n = 30 cells from three independent experiments).

in aneuploidy is triggered by loss of Wnt signaling and is mediated by an abnormal increase in microtubule dynamics in mitotic spindles. This causes the generation of erroneous, merotelic microtubule-kinetochore attachments leading to the formation of lagging chromosomes in anaphase as a pre-stage of chromosome missegregation. It is currently not clear on a mechanistic level how abnormal microtubule dynamics is translated into faulty microtubule-kinetochore attachments, but previous work from our laboratory has shown that increased microtubule growth rates cause transient spindle misorientation in the early phases of mitosis (30). This might generate a condition where transient spindle geometry changes facilitate the generation of erroneous, merotelic microtubule-kinetochore attachments that overwhelm the cellular correction machinery that otherwise resolves such erroneous kinetochore attachments

IWP12 for 24 h. Scatter dot plots show average assembly rates (20 microtubules/cell, mean ± SEM, $t$ test, n = 30 cells from three independent experiments). **(F)** Proportion of IWP12-treated HCT116 cells exhibiting lagging chromosomes. Cells were treated as in (E), and the graph shows mean values ±SEM ($t$ test, n = 300 cells from three independent experiments).

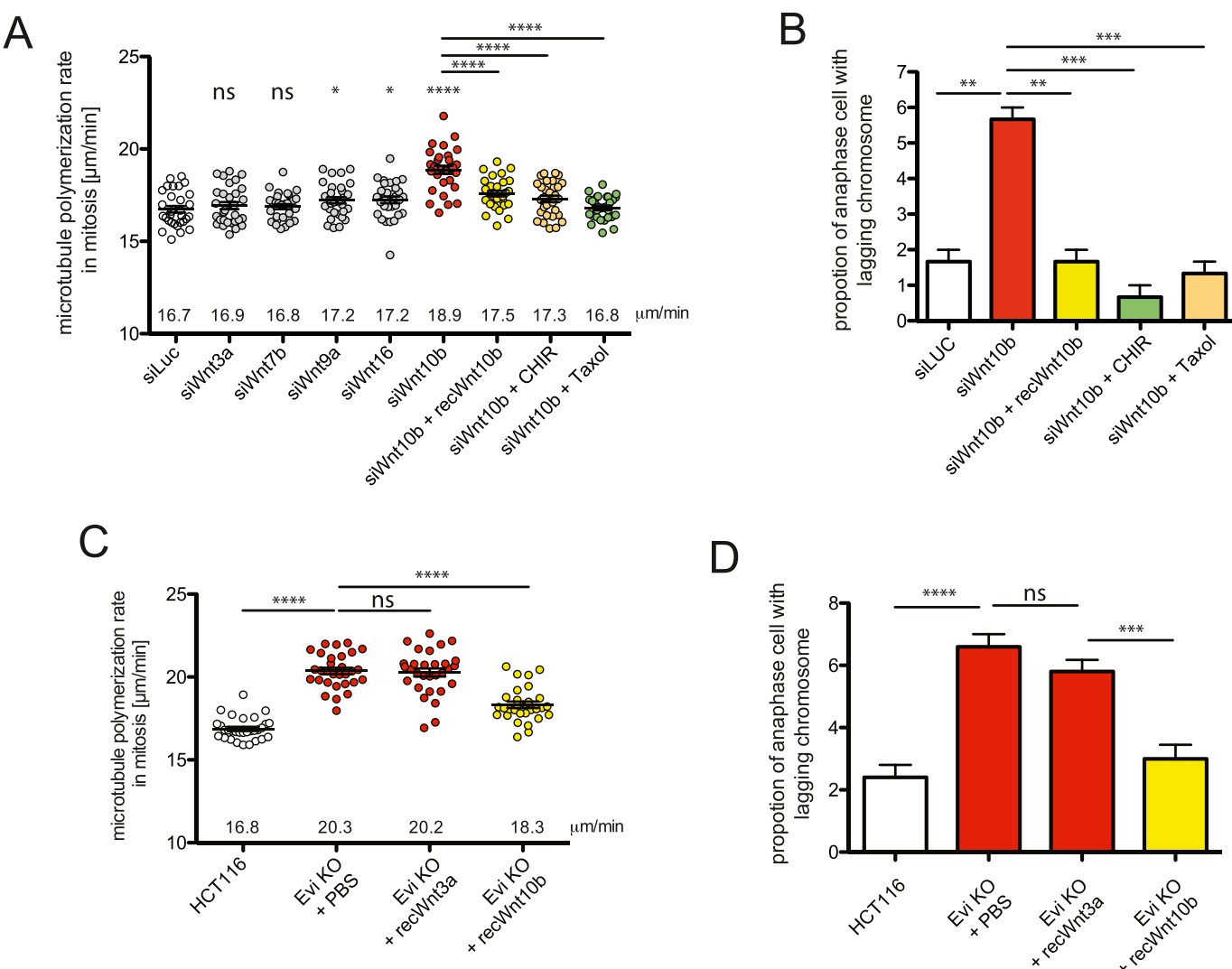

**Figure 5. Basal Wnt10b signaling is required for proper mitotic regulation.**
**(A)** Mitotic microtubule plus-end assembly rates in HCT116 cells after siRNA-mediated down-regulation of different Wnts. Cells were transfected with siRNAs targeting the indicated Wnts and subjected to microtubule plus end growth measurements in live cells. 24 h post transfection, the cells were additionally treated with 400 ng/ml recombinant Wnt10b protein, 0.6 $\mu$M CHIR99021, or 0.2 nM Taxol for another 24 h before measurements. Scatter dot plots show average assembly rates (20 microtubules/cell, mean ± SEM, $t$ test, n = 30 cells from three independent experiments). **(B)** Proportion of cells exhibiting lagging chromosomes. **(A)** HCT116 cells were transfected with siRNAs targeting Wnt10b and treated with Taxol, CHIR99021, or recombinant Wnt10b protein as in (A). The graph shows mean values ±SEM ($t$ test, n = 300 cells from three independent experiments). **(C)** Mitotic microtubule plus end assembly rates in HCT116$^{Evi\_KO}$ cells after treatment with 400 ng/ml recombinant Wnt3a or Wnt10b protein for 16 h. Scatter dot plots show average assembly rates (20 microtubules/cell, mean ± SEM, $t$ test, n = 30 cells from three independent experiments). **(D)** Proportion of HCT116$^{Evi\_KO}$ cells exhibiting lagging chromosomes after treatment with Wnt3a or Wnt10b. **(C)** HCT116$^{Evi\_KO}$ cells were treated with recombinant Wnt3a or Wnt10b protein as in (C) and the proportion of cells exhibiting lagging chromosomes were determined by immunofluorescence microscopy. The graph shows mean values ±SEM ($t$ test, n = 500 cells from five independent experiments).

(27). Interestingly, mitotic spindle misorientation has also been described after depletion of DVL2 in HeLa cells (36), but whether this was also associated with increased microtubule dynamics was not addressed. Our own previous data, however, showed that siRNA-mediated depletion of DVL2 was sufficient to increase microtubule dynamics in mitosis (26).

Here, we further defined the mitosis and chromosome segregation relevant Wnt signaling pathway. It involves the components of the "canonical pathway" comprising Frizzled receptors, LRP5/6 co-receptors, DVLs and the destruction complex, but not $\beta$-catenin.

Thus, Wnt-mediated stabilization of proteins other than $\beta$-catenin (Wnt/STOP) seems to be required for proper mitosis (10, 18, 19, 20, 21). In fact, previous work has established that Wnt signaling down-regulates GSK3$\beta$ activity, thereby preventing phosphorylation of proteins that are otherwise targeted for degradation (10, 37). This is fully in line with our results showing that increased mitotic microtubule polymerization rates and chromosome missegregation after inhibition of Wnt signaling fully depends on GSK3$\beta$ activity. Moreover, a moderate increase in GSK3$\beta$ activity is sufficient to trigger the same mitotic defects as seen after inhibition of Wnt

signaling. Hence, an important function of Wnt signaling in this context seems to be a constant down-regulation of a sub-pool of Wnt-controlled basal GSK3$\beta$ activity. It is interesting that fine-tuned regulation of Wnt associated GSK3$\beta$ activity seems not only be important for mitotic regulation, but also, for example, for mac-ropinocytosis and metabolite changes as demonstrated recently (38). Importantly, down-regulation of basal GSK3$\beta$ required for proper mitosis is mediated by basal Wnt signaling that requires ongoing secretion of Wnt ligands. In fact, our work showed that PORCN inhibition or knockout of the Wnt cargo receptor Evi/Wls is sufficient to trigger the mitotic defects leading to chromosome missegregation in a GSK3$\beta$-dependent manner. Thus, basal Wnt signaling represents a self-maintaining machinery required for proper mitosis and genome stability. This is in full agreement with a previous study in mESCs with Evi/Wls knockout where loss of Wnt secretion is also associated with mitotic abnormalities and genome instability (17). Because of technical limitations, we were not able to test whether increased microtubule polymerization is also involved in genome instability in these mESCs. Interestingly, this previous work showed that mitotic abnormalities in Evi/Wls knockout mESCs could be rescued by re-addition of Wnt3a and Wnt5 (17). In contrast, our data in somatic cells clearly indicate that Wnt3a or Wnt5 are not able to rescue the abnormally increased microtubule growth rates and chromosome missegregation, which might indicate that dif-ferent Wnts might contribute to mitotic regulation, depending on the cell type. In fact, we performed a comprehensive screen of 14 different human Wnts for their involvement in the regulation of microtubule dynamics in mitosis. We identified Wnt10b as a major Wnt ligand regulating mitotic microtubule dynamics, at least in HCT116 and in non-transformed RPE1 cells. In addition to that, also Wnt7b and Wnt9a might be candidate ligands with functions for mitosis, possibly in other cell types and/or cell lines. It is intriguing that only a small subset or even one specific Wnt ligand regulates mitotic microtubule dynamics through the "canonical" pathway and by restraining GSK3$\beta$ activity. It has been reported that Wnt10b acts as a "canonical" Wnt, but with lower activity in $\beta$-catenin reporter assays when compared with, for example, Wnt3a ((3) and our data). This might reflect that strong canonical Wnts (e.g., Wnt3a), which induced signaling through a broad spectrum of FZD receptors (7), might preferentially regulate $\beta$-catenin stability. Other Wnts, in-cluding Wnt10b, which act through a more restricted set of FZDs (7), might use the same pathway to mediate Wnt/STOP and thus, stability of proteins other than $\beta$-catenin. It will be important for future work to dissect separable physiological function of different Wnts using the same signaling pathway. To our knowledge, the finding that Wnt10b fulfills an essential function for mitosis is the first example of a specific Wnt ligand required for normal cell cycle regulation in human somatic cells. Wnt10b is expressed during embryogenesis and also in adult tissues, but its role has not been well studied in somatic cells. Studies using transgenic and knockout mice, however, demonstrated a role for Wnt10b in the immune system, in the development of the mammary gland and the reg-ulation of bone mass (39). It is interesting to note that Wnt10b expression was found to be down-regulated in the aging brain and in Alzheimer disease brains (40), conditions that are known to be associated with aneuploidy. Intriguingly, down-regulation of the Wnt components DVL and LRP6 and up-regulation of the Wnt/LRP6

antagonist DKK1 was also found in aging brains (40). Based on our work, all these conditions are sufficient to induce aneuploidy by preventing the faithful execution of mitosis, possibly providing a link between loss of Wnt10b-LRP6-GSK3$\beta$ signaling and the in-duction of aneuploidy in human age-related disease conditions.

The link between Wnt signaling and aneuploidy might also have clinical implications. It is well established that hyper-active Wnt signaling has a tumor-promoting function (37). Therefore, inhibition of Wnt signaling has been considered as an anti-cancer treatment strategy. In fact, several PORCN inhibitors, FZD antagonists, secreted frizzled-related proteins, or tankyrase inhibitors are currently tested in clinical phase1/2 trials for the treatment of various cancers (37). Based on our results, these treatments would induce chromosomal instability and aneuploidy, conditions known to contribute to tumor progression by supporting tumor evolution and by facilitating the development of resistance towards chemo-therapy (41). This would put Wnt inhibitors at risk for the treatment of cancer and might even promote tumor progression. On the other hand, therapies aiming to activate Wnt signaling, including DKK-neutralizing antibodies or GSK3$\beta$ kinase inhibitors, have been considered for treatment of diseases with reduced Wnt signaling such as Alzheimer's disease (40, 42). Here, evolving aneuploidy, which is associated with such neurodegenerative diseases (43), might be indeed suppressed by Wnt activators as we have dem-onstrated in our cell-based studies.

# Materials and Methods

### Cell culture and treatments

HCT116 cells were obtained from ATCC and grown in RPMI1640 medium supplemented with 10% FBS and 100 $\mu$g/ml streptomycin and 10 U/ml penicillin (Invitrogen). RPE-1-hTert cells (a kind gift of Zuzana Storchova, University of Kaiserslautern) were grown in DMEM/F12 medium supplemented with 10% FBS, 100 $\mu$g/ml strep-tomycin and 10 U/ml penicillin (Invitrogen). HEK293T cells were a kind gift from Julai Gross (University Medical Center Göttingen). All cells were cultured at 37°C under 5% $CO_2$. Where indicated, the cells were treated with 600 ng/ml Dickkopf1 (DKK1, PeproTech), 400 ng/ml Wnt3a (R&D Systems), 400 ng/ml Wnt10b (R&D Systems), 0.6 $\mu$M CHIR99021 (Sigma-Aldrich), and 5 $\mu$M IWP12 (Sigma-Aldrich).

### Generation of Evi/Wls knockout cells

HCT116^Evi_KO cells were generated using CRISPR/Cas9 following a protocol described in reference 44. In short, the HCT116 cells were transfected with px459 vector (a gift from Feng Zhang, Addgene plasmid # 48139; http://n2t.net/addgene:48139; RRID: Addgene_48139) expressing Cas9 and sgRNA targeting human Evi/Wls (5′-GTAAGC-CAGGGAAACGTCCA-3′). After 48 h, the cells were selected with 2 $\mu$g/ml of puromycin (Invitrogen) for 72 h. Because Evi silencing in HCT116 cells resulted in growth inhibition (16), either recombinant Wnt3a (50 ng/ml; R&D Systems) and/or medium from parental cell lines was added to the growth medium. Single-cell clones were selected and were used

for further analysis. The generation of HCT116[Evi_KO] cells will be described in detail in Voloshanenko et al (manuscript in preparation).

## Generation of stable cell lines

To generate stably expressing GSK3β, HCT116 cells were transfected with pCDNA3-GSK3β-HA (a gift from Jim Woodgett, Addgene plasmid #14753; http://n2t.net/addgene:14753; RRID: Addgene_14753addgene) using Lipofectamine 3000 (Thermo Fisher Scientific) according to the instructions of the manufacturer. Single-cell clones were selected in RPMI medium supplemented with 300 mg/ml G418 (Invitrogen) and further analyzed.

## siRNA transfection

Cells were transfected with 40–80 pmol siRNAs (Sigma-Aldrich) using Lipofectamine 3000 (Thermo Fisher Scientific) or ScreenFectsiRNA (ScreenFect GmbH). After 48–72 h, the cells were further analyzed. The following siRNAs were used: luciferase, LUC: 5′-CUUACGCUGAGUA-CUUCGAUU-3′. Wnt3a: 5′-GCCAUGAACCGCCACAACAtt-3′. Wnt7b 5′-GCAG-GAAGGUUCUAGAGGAtt-3′. Wnt9a: 5′-GGUGCGUUGGUGCUGCUAUtt-3′. Wnt10b: 5′-CCACAACCGCAAUUCUGGAtt-3′. Wnt16: 5′-GGCAGAGAA-UGCAACCGUAtt-3′.

## Generation of Wnt-conditioned media

Wnt-conditioned media were generated according to standard procedures. Briefly, HEK293T cells were transfected using Lipofectamine 3000 (Thermo Fisher Scientific) according to the manufacturer protocol. 2.25 μg of untagged pcDNA3-Wnt plasmids (Addgene, Open Source Wnt Project Plasmids, Kit #1000000022) (3) and 0.25 μg pEGFP (to test for equal transfection efficiency) were used for transfection of cells in six-well format. 48 h post transfection, fresh medium was added, and conditioned medium was collected after 72 h.

## FACS

FACS analysis was performed using a FACS Canto II flow cytometer (Beckton Dickinson) as described previously (30). To detect apoptotic cells, entire cell populations including floating cells were harvested and stained with 5 μg/ml propidium iodide. Cells with a sub-G1 DNA content were quantified as apoptotic cells using BD FACSDIVA software.

## Live cell microscopy

HCT116 stably expressing H2B-GFP cells were grown in RPMI1640 medium without phenol red supplemented with 10% FBS and seeded into eight-well glass-bottom dishes (Ibidi). Live-imaging was performed using a Deltavision ELITE microscope (GE Healthcare) equipped with an Olympus x60 1.42 NA objective and a PCO Edge sCMOS camera (PCO). Images were recorded every 5 min for up to 5 h, whereas cells were incubated at 37°C and 5% of $CO_2$. Images were deconvolved using SoftWorx 5.0/6.0 software (Applied Precision).

## Determination of microtubule plus-end assembly rates

Microtubule plus-end growth rates were determined by tracking Eb3-GFP in living cells as described (30). Cells were transfected 48 h before the measurement with pEGFP-EB3 (kindly provided by L Wordeman, University of Washington), seeded onto glass-bottom dishes (Ibidi) and treated with Dimethylenastron (DME, 2 μM, Calbiochem) for 2 h before measurements. Live-imaging was performed using a Deltavision ELITE microscope (GE Healthcare) equipped with an Olympus x60 1.42 NA objective and a PCO Edge sCMOS camera (PCO) and images were recorded every 2 s, whereas cells were incubated at 37°C and 5% of $CO_2$. Images were deconvolved using SoftWorx 5.0/6.0 software (Applied Precision) and average assembly rates were calculated for 20 individual microtubules per cell. 30 cells were analyzed in three independent experiments. To restore proper microtubule plus-end growth rates, 0.2 nM Taxol (Sigma-Aldrich) or 0.6 μM CHIR99021 (Sigma-Aldrich) was used.

## Analysis of lagging chromosomes in anaphase

To enrich cells in anaphase, the cells were treated with 2 mM thymidine for 20 h followed by release into fresh medium for 9 h. The cells were fixed in methanol at −20°C and analyzed by immunofluorescence microscopy. Mitotic spindles, chromosomes, and centromeres were detected using anti-α-tubulin antibodies (1:100; Santa Cruz), Hoechst 33342 (1:20,000; Biomol), and anti-CENP-C antibodies (1:1,000; MBL), respectively. Only CENP-C-positive chromosomes clearly separated from two masses of chromosomes were defined as lagging chromosomes.

## Reporter assays

HCT116 cells were seeded into 96-well plates and transfected with 10 ng Topflash luciferase and 5 ng Renilla luciferase reporter plasmid using XtremeGene9 DNA transfection reagent (Roche). 24 h later, cells were treated with PBS, 400 ng/ml Wnt10b or 400 ng/ml Wnt3a. After 16 h, the cells were lysed with commercial Passive Lysis Buffer (Promega) for 15 min at 4°C. 30 μl of lysate was transferred to a white 96-well plate and analyzed using a Dual-Luciferase Reporter Assay System (Promega) using a Tecan Microplate Reader (Tecan Infinite M1000). Topflash signals were normalized to the Renilla reporter and Wnt activity was calculated relative to the control condition.

## qRT-PCR analysis

Total cellular RNAs were isolated using peqGold TriFast reagent (Peqlab) according to the manufacturer's instructions. The concentrations of RNA were determined using a NanoDrop ND-2000 (NanoDrop). The first-strand cDNA synthesis using total cellular RNAs as the template was performed with Invitrogen RT Reagent Kit. qRT-PCR was performed with iTaq Universal SYBR Green Supermix (Bio-Rad Laboratories GmbH) as the dsDNA fluorescence dye using CFX96 Real-Time PCR Detection Systems (Bio-Rad Laboratories, Inc.). The reactions were performed under the following conditions: 95°C for 3 min; 40 cycles of 95°C for 10 s, 60°C for 30 s, and 72°C for

30 s; and a melting curve from 65°C to 95°C. The results of qRT-PCR were defined from the threshold cycle (Ct), and relative mRNA levels were determined using the $2^{-\triangle\triangle Ct}$ method with beta-actin ($\beta$-actin) as an internal control. The following primer pairs used for qRT-PCR:

wnt3a: 5′-CCTGCACTCCATCCAGCTACA-3′ and
5′-GACCTCTCTTCCTACCTTTCCCTTA-3′;
wnt7b: 5′-CCCGGCAAGTTCTCTTTCTTC-3′ and
5′-GGCGTAGCTTTTCTGTGTCCAT-3′;
Wnt9a: 5′-TGCCTTCCTCTATGCCATCTCC-3′and
5′-TCCTTGACGAACTTGCTGCTG-3′;
Wnt10b: 5′-GAGTGCAAGGTTACAGAGTGGG-3′ and
5′-TCAGAGCAAAGGGCTGAAAAGG-3′;
Wnt16: 5′-CTACAGCTCCCTGCAAACGA-3′;
$\beta$-actin: 5′-GAGCACAGAGCCTCGCCTTT-3′and
5′-ACATGCCGGAGCCGTTGTC-3′.

### Karyotype analysis

To determine chromosome number variability evolved over time, single-cell clones were generated and grown for 30 generations. To enrich for mitotic cells, the cells were treated with 2 μM Dimethylenastron (Sigma-Aldrich) for 4 h. The cells were collected and incubated in hypotonic medium (40% RPMI-1640, 60% $H_2O$) at RT for 15 min. The cells were fixed in Carnoy's solution (methanol:acetic acid = 3:1). Chromosomes were spread onto glass slides and stained with Giemsa solution (Sigma-Aldrich). The proportion of cells with chromosome numbers deviating from the modal was determined by chromosome counting.

### Western blotting

Cells were lysed in lysis buffer (50 mM Tris–HCl, pH 7.4, 150 mM NaCl, 5 mM EDTA, 5 mM EGTA, 1% [vol/vol] NP-40, 0.1% [wt/vol] SDS, 0.1% sodium desoxycholate, protease inhibitor cocktail [Roche], and phosphatase inhibitor cocktail [Roche]). Proteins were resolved on 7.5%, 10%, or 12% SDS polyacrylamide gels and blotted onto nitrocellulose membranes (Roth) using semi-dry or tank-blot procedures. For Western blot experiments, the following antibodies and dilutions were used: anti-$\beta$-actin (1:40,000, AC-15; Sigma-Aldrich), anti-GSK3$\beta$ (1:5,000, 27C10; Cell Signaling), anti-Evi (1:500, YJ5; BioLegend), and anti-$\alpha$-tubulin antibodies (1:5,000; Santa Cruz).

### Statistical analysis

For all data, mean values and SEM were calculated using GraphPad Prism 5.0 software (Graph Pad Software). Statistical analysis was performed using two-sided unpaired $t$ tests and significances are indicated as:

* = $0.01 < P < 0.05$; ** = $0.001 < P < 0.01$; *** = $P < 0.001$, **** = $P < 0.0001$.

## Data Availability

This study includes no data deposited in external repositories.

## Supplementary Information

## Acknowledgements

We thank Jim Woodgett, Feng Zhang, Linda Wordeman, and Zuzana Storchova for plasmids and cell lines. This work was funded by the Deutsche Forschungsgemeinschaft (DFG, German Research Foundation) Project No. 331351713 – SFB 1324 "Mechanisms and Functions of Wnt signaling" (H Bastians, SP Acebron, J Gross, M Boutros).

### Author Contributions

Y-C Lin: conceptualization, data curation, formal analysis, investigation, methodology, and writing—original draft, review, and editing.
A Haas: formal analysis, investigation, methodology, and writing—review and editing.
A Bufe: formal analysis, investigation, and methodology.
S Parbin: formal analysis, investigation, and methodology.
M Hennecke: formal analysis, investigation, and methodology.
O Voloshanenko: formal analysis, investigation, and methodology.
J Gross: conceptualization, investigation, and methodology.
M Boutros: conceptualization, formal analysis, and methodology.
SP Acebron: conceptualization, data curation, formal analysis, supervision, investigation, methodology, and writing—review and editing.
H Bastians: conceptualization, data curation, formal analysis, funding acquisition, project administration, and writing—original draft, review, and editing.

### Conflict of Interest Statement

The authors declare that they have no conflict of interest.

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
