## [Reviewer comments · Life Science Alliance]

Life Science Alliance

Wnt10b-GSK3 β -dependent Wnt/STOP signaling prevents aneuploidy in human somatic cells

Yu-Chih Lin, Alexander Haas, Anja Bufe, Sabnam Parbin, Magdalena Hennecke, Oksana Voloshanenko, Julia Gross, Michael Boutros, Sergio Acebron, and Holger Bastians

DOI: <https://doi.org/10.26508/lsa.202000855>

Corresponding author(s): Holger Bastians, University Medical Center Goettingen

Review Timeline:

Submission Date:	2020-07-22
Editorial Decision:	2020-08-20
Revision Received:	2020-11-09
Editorial Decision:	2020-11-15
Revision Received:	2020-11-16
Accepted:	2020-11-17

Scientific Editor: Shachi Bhatt

Transaction Report:

August 20, 2020

Re: Life Science Alliance manuscript #LSA-2020-00855-T

Prof. Holger Bastians
University Medical Center Goettingen
GZMB Molecular Oncology
Grisebachstrasse 8
Göttingen 37077
Germany

Dear Dr. Bastians,

Thank you for submitting your manuscript entitled "Wnt10b-dependent signaling is required for proper mitosis and to prevent aneuploidy in human somatic cells" to Life Science Alliance (LSA). The manuscript has been reviewed by the editors and outside referees (reviewer comments below). As you will see, the reviewers were enthusiastic about the study, but have raised several concerns that should be addressed prior to further consideration of the manuscript at LSA. Therefore, although we are unable to publish the current version of the manuscript, we would encourage you to submit a revised version that addresses the referees' concerns, including the concerns raised by Rev 1 about why other Wnts that reportedly act via the LRP6/Dvl/GSK axis do not rescue in a similar fashion to Wnt10b in this system, OR data testing the potential of other Wnts to rescue microtubule polymerization in other cells, to see if there was a sub-group of Wnt family that does similar functions in different cells types could be interesting too. All other minor concerns raised by both Rev 1 and Rev 2 should also be addressed in the revised manuscript.

We would be happy to discuss the individual revision points further with you should this be helpful. The typical timeframe for revisions is three months. Please note that papers are generally considered through only one revision cycle, so strong support from the referees on the revised version is needed for acceptance. While you are revising your manuscript, please also attend to the below editorial points to help expedite the publication of your manuscript. Please direct any editorial questions to the journal office.

We hope that the comments below will prove constructive as your work progresses. Thank you for this interesting contribution to Life Science Alliance. We are looking forward to receiving your revised manuscript.

Sincerely,

Shachi Bhatt
Executive Editor
Life Science Alliance

B. MANUSCRIPT ORGANIZATION AND FORMATTING:

Reviewer #1 (Comments to the Authors (Required)):

The Bastians group has previously shown that inhibition of endogenous Wnt signaling by knockdown of LRP6 and Dvl2 as well as DKK1 treatment to block LRP6 function leads to increased microtubule assembly rates, abnormal mitotic spindle formation, and aneuploidie in HCT116 colon tumor cells (Stolz 2015b). They extend this analysis here by showing that inhibition of GSK3 rescues these effects and that overexpression of GSK3 suffices to induce these effects, in line with the known negative downstream role of GSK3 in canonical Wnt signaling. Moreover, blocking secretion of Wnts yielded similar mitotic defects as interference with LRP6 receptors in line with an autocrine

mechanism of Wnt signaling in these cells.

My main concern with this manuscript is that results, although well worked out and presented, do not go much beyond what was already shown in the earlier paper. Most experimental approaches mainly strengthen the view that an autocrine Wnt mechanism operates in controlling mitotic processes. That Wnt10b, in contrast to other Wnts expressed in HCT116 is specifically required is of interest, but no attempt is made to dissect why the other Wnts that are also capable of activating Wnt signaling don't work. Moreover, we are not closer to understanding how the Wnt/LRP6/GSK cascade affects microtubule assembly.

On the other hand, as the study in general conforms with the LSA requirements that welcome confirmatory studies that advance the field, if there are no technical limitations, the verdict might be less strict, provided some specific critical aspects are addressed in a revised version.

Specific comments

1 In Figure 1 GSK3 is inhibited by CHIR99021 treatment. It is not clear from the paper how long this treatment was performed. In Stolz 2015b, DKK1 was added for 24 hrs. Assuming that this is the case for CHIR99021 treatment here as well, was cell viability altered by such a long inhibition of GSK3, which is a key kinase of several pathways? How long was the treatment for karyotype analysis, which took place after 30 cell generations? The authors should control for cell viability using appropriate assays, because differences in cell survival could skew results, for instance if cells with mitotic abnormalities plus inhibited GSK3 die preferentially.

2 As mentioned above, the finding that only knockdown of Wnt10b, but not of the other Wnts altered microtubule assembly and mitotic behavior, and that only Wnt10b but not Wnt3a could rescue Evi KO is of particular interest. It is difficult to understand why Wnts that reportedly act via the LRP6/Dvl/GSK axis do not rescue.

Of note, Wnt3a and Wnt5 rescued mitotic defects after EviKo in mouse embryonic stem cells as reported in the Augustin, 2017 paper from the Boutros lab. It would be interesting if Wnt10b would be active here as well. Also, it would be interesting if alterations in microtubule assembly were observed in these cells.

To make their analysis more comprehensive, I suggest that the authors test a few more Wnts (best all of the known ones) for rescue activity to figure out whether there are others with similar activity as Wnt10b, which could define a subgroup of Wnts, maybe active in other cells. Test for rescue of microtubule polymerization should suffice as a readout.

Minor

Some references are not complete.

Please give details of times of treatment for the various settings.

Reviewer #2 (Comments to the Authors (Required)):

Lin et al. Wnt10b-dependent signaling is required for proper mitosis....

This is a very original paper that continues the saga of Wnt/STOP cellular regulation independently of beta-catenin. I have several criticisms that could be readily addressed considering the expertise of Dr. Acebron, the father of Wnt/STOP, in this field.

Major comments:

1. Almost all the work is based on the use of HCT116 colon carcinoma cells. This is fine as they control with euploid cells too, but the reader is not told that HCT116 cells cause cancer through a stabilizing mutation of beta-catenin. This is why they have high TCF4/Wnt reporter activity.
 2. What I find amazing is that cells with stabilized beta-catenin are still dependent on Dkk, Lrp6 and GSK3. This must be due to Wnt/STOP of mitotic proteins, and has to be explained to the reader. Wnt/STOP is not even mentioned in the abstract or title.
 3. The remarkable finding is that basal GSK3 activity is regulated by Dkk/Lrp6, and as DKK decreases GSK3 sequestration in endosomes, the increase in GSK3 activity in the cytosol causes the cytogenetic abnormalities. That seems to be the main take home lesson of the paper: GSK3 activity is constantly downregulated by LRP6/Wnt-Porcupine signaling. The authors highlight the role of Wnt10b at the expense of the basal cellular activity of GSK3. This is a mistake in my view, because in these particular cells Wnt10b might be special but in others it will be other Wnts. Wnt10b is crucial in mammary gland and not that many other tissues, while GSK3 is universal. The title should mention GSK3 and Wnt/STOP in some way.
 4. An example of the assay of EB3 imaging to determine microtubule growth rates should be shown in the context of this paper in the EV supplemental information. The reader cannot be expected to refer back to the Stoltz et al. 2015, as is requested repeatedly. The quantitations are beautiful but some EV images are needed as well.
- Minor Comments:
5. page 5 line 28, change demonstrating to suggesting. It is not demonstrated here.
 6. page 9 lines 17-21 and Fig. 4C do not seem useful. Other cells will respond to other Wnts but in the end it is the increase in cytosolic GSK3 what counts.
 7. page 9, last paragraph, great experiment. How does one identify the mutant cells from the excess of parental cells?
 8. In a paper just published, Albrecht et al. Cell Reports 2020, shows that Axin1 mutation or GSK3 inhibition triggers macropinocytosis and metabolite changes in a way independent of protein synthesis and beta-catenin stabilization. Axin1, APC and GSK3 repress macropinocytosis in basal cellular conditions. The basal levels of GSK3 activity seem to be very important in cell physiology.
 9. The authors should discuss what would be the predicted effects of Dkk, Dkk antibodies or Porcupine inhibitors in a clinical setting as these seem to be promising treatments.
 10. It is surprising that Davidson et al. Dev Cell 2009 is not even cited, for this study showed that Wnt signaling is elevated during the G2 and M phases of the cell cycle.
 11. Please say when first introduced that EV means Expanded View. I did not know.

Rebuttal letter**Lin et al., manuscript #LSA-2020-00855-T****Original title: "Wnt10b-dependent signaling is required for proper mitosis and to prevent aneuploidy in human somatic cells"****New title: "Wnt10b-GSK3 β -dependent Wnt/STOP signaling prevents aneuploidy in human somatic cells."****Reviewer #1**

The Bastians group has previously shown that inhibition of endogenous Wnt signaling by knockdown of LRP6 and Dvl2 as well as DKK1 treatment to block LRP6 function leads to increased microtubule assembly rates, abnormal mitotic spindle formation, and aneuploidy in HCT116 colon tumor cells (Stolz 2015b). They extend this analysis here by showing that inhibition of GSK3 recues these effects and that overexpression of GSK3 suffices to induce these effects, in line with the known negative downstream role of GSK3 in canonical Wnt signaling. Moreover, blocking secretion of Wnts yielded similar mitotic defects as interference with LRP6 receptors in line with an autocrine mechanism of Wnt signaling in these cells.

My main concern with this manuscript is that results, although well worked out and presented, do not go much beyond what was already shown in the earlier paper. Most experimental approaches mainly strengthen the view that an autocrine Wnt mechanism operates in controlling mitotic processes. That Wnt10b, in contrast to other Wnts expressed in HCT116 is specifically required is of interest, but no attempt is made to dissect why the other Wnts that are also capable of activating Wnt signaling don't work. Moreover, we are not closer to understanding how the Wnt/LRP6/GSK cascade affects microtubule assembly.

On the other hand, as the study in general conforms with the LSA requirements that welcome confirmatory studies that advance the field, if there are no technical limitations, the verdict might be less strict, provided some specific critical aspects are addressed in a revised version.

We thank the reviewer for his critical comments. We realize that our work is not providing an in-depth mechanistic explanation of how Wnt affects microtubule dynamics and chromosome segregation in mitosis. However, we still think that our data are of interest to the field and would deserve publication in LSA. In fact, we demonstrate that self-maintaining Wnt signaling, which requires constant secretion of Wnts and constant down-regulation of GSK3 activity is required for proper regulation of mitosis. We find it also intriguing that, although several Wnts can activate the "canonical" pathway, only a few (maybe only wnt10b) are involved in this particular mitotic regulation. Of course, we are very much interested in following up on our observations presented here and we clearly aim to further dissect the role of the different Wnt ligands in the context of mitotic regulation.

As the reviewer suggested, we performed additional experiments to make our

study more comprehensive. These are detailed below.

Specific comments:

1 In Figure 1 GSK3 is inhibited by CHIR99021 treatment. It is not clear from the paper how long this treatment was performed. In Stolz 2015b, DKK1 was added for 24 hrs. Assuming that this is the case for CHIR99021 treatment here as well, was cell viability altered by such a long inhibition of GSK3, which is a key kinase of several pathways? How long was the treatment for karyotype analysis, which took place after 30 cell generations? The authors should control for cell viability using appropriate assays, because differences in cell survival could skew results, for instance if cells with mitotic abnormalities plus inhibited GSK3 die preferentially.

We used CHIR99021 for only 16 hours when determining microtubule dynamics and chromosome missegregation. Unfortunately, we omitted this information in the text. The treatment times are now provided for all experiments in the figure legends.

It is of note that we used only low concentrations of the inhibitor (0.6 μ M) and we have not noted any cell death in response to these inhibitor treatments. However, we are grateful for this comment and, as the reviewer suggested, we have performed FACS analyses, which can detect apoptosis/cell death and also cell cycle abnormalities in response to the treatments. By using the low concentration of CHIR99021 we could neither detect cell cycle changes nor cell death. These new data are now shown in the new Supplemental Figure 1C and described on page 5: *“Rigorous quantification of the proportion of cells exhibiting lagging chromosomes indeed revealed a significant induction of lagging chromosomes upon 16 hour DKK1 treatment, which was suppressed by co-treatment with a low concentration of CHIR99021 (Fig. 1A), which did not affect cell viability (Supplemental Fig. 1C).”*

2. As mentioned above, the finding that only knockdown of Wnt10b, but not of the other Wnts altered microtubule assembly and mitotic behavior, and that only Wnt10b but not Wnt3a could rescue Evi KO is of particular interest. It is difficult to understand why Wnts that reportedly act via the LRP6/Dvl/GSK axis do not rescue. Of note, Wnt3a and Wnt5 rescued mitotic defects after EviKo in mouse embryonic stem cells as reported in the Augustin, 2017 paper from the Boutros lab. It would be interesting if Wnt10b would be active here as well. Also, it would be interesting if alterations in microtubule assembly were observed in these cells.

We were also surprised to see that neither Wnt3a nor Wnt5a were able to suppress the mitotic defects in somatic cells. However, stem cells and somatic cells might behave differently. As the reviewer mentioned, it is currently not clear whether Evi/Wls knockout mECSs exhibit increased microtubule polymerization rates. In fact, we collaborated already previously with the lab of M. Boutros to address this question (as part of the paper submission by Augustin et al). Unfortunately, although we have long-standing experience in measuring microtubule dynamics in living cells, we were unable to determine microtubule polymerization rates in a reliable manner. This was due to technical limitations:

mitotic (rounded) mESCs do not properly attach to glass surfaces, which is associated with movement of cells during the microscopic measurements. Given the fact that we detect very small signals at individual microtubule plus tips this excluded proper calculation of microtubule growth rates in mESCs. The use of strategies to artificially tether cells to glass surfaces are not suitable due to the fact that interference with integrin-mediated cell adhesion grossly affects microtubule polymerization in mitosis.

Our rescue results indicating the differences between mESC and somatic cells are now discussion in the new Discussion section: page 11: *“Thus, basal Wnt signaling represents a self-maintaining machinery required for proper mitosis and genome stability. This is in full agreement with a previous study in mouse embryonic stem cells (mESCs) with Evi/Wls knockout where loss of Wnt secretion is also associated with mitotic abnormalities and genome instability [14]. Due to technical limitations we were not able to test whether increased microtubule polymerization is also involved in genome instability in these mESCs. Interestingly, this previous work showed that mitotic abnormalities in Evi/Wls knockout mESCs could be rescued by re-addition of Wnt3a and Wnt5 [14]. In contrast, our data in somatic cells clearly indicate that Wnt3a or Wnt5 are not able to rescue the abnormally increased microtubule growth rates and chromosome missegregation, which might indicate that different Wnts might contribute to mitotic regulation, depending on the cell type.”*

To make their analysis more comprehensive, I suggest that the authors test a few more Wnts (best all of the known ones) for rescue activity to figure out whether there are others with similar activity as Wnt10b, which could define a subgroup of Wnts, maybe active in other cells. Test for rescue of microtubule polymerization should suffice as a readout.

This is an excellent suggestion. We have now performed a comprehensive analysis of Wnts involved in the regulation of mitotic microtubule polymerization. For this we used HCT116-Evi knockout cells and treated them with conditioned media for 14 different Wnts. Subsequent systematic measurements of microtubule dynamics revealed that only Wnt10b and, to a lesser extent, Wnt7b and Wnt9a, can rescue the abnormally increased microtubule polymerization rates in Evi knockout cells. The results of this screening approach are now presented in our new Figure 4B.

It is interesting to note that downregulation of Wnt10b, but not of Wnt7a or Wnt9b, is sufficient to induce abnormal microtubule dynamics in HCT116 cells (Fig. 5A). This might indicate, as the reviewer supposed, that Wnt10b is indeed the most relevant Wnt in HCT116 cells and additional Wnts like Wnt7a or Wnt9b might be more relevant in other cell types / cell lines.

We discuss this possibility in the new Discussion section, page 11: *“In fact, we performed a comprehensive screen of 14 different human for their involvement in the regulation of microtubule dynamics in mitosis. We identified Wnt10b as a major Wnt ligand regulating mitotic microtubule dynamics, at least in HCT116 and in non-transformed RPE1 cells. In addition to that, also Wnt7b and Wnt9a might be candidate ligands with functions for mitosis, possibly in other cell types and/or cell lines.”*

Minor

Some references are not complete.

We carefully re-checked all references and realized that for some reference there are no page numbers available. Instead manuscript numbers are provided.

Please give details of times of treatment for the various settings.

We have now included the treatment times and all concentrations of inhibitors etc. for each experiment in the figure legends.

Reviewer #2

Lin et al. Wnt10b-dependent signaling is required for proper mitosis....

This is a very original paper that continues the saga of Wnt/STOP cellular regulation independently of beta-catenin. I have several criticisms that could be readily addressed considering the expertise of Dr. Acebron, the father of Wnt/STOP, in this field.

Major comments:

1. Almost all the work is based on the use of HCT116 colon carcinoma cells. This is fine as they control with euploid cells too, but the reader is not told that HCT116 cells cause cancer through a stabilizing mutation of beta-catenin. This is why they have high TCF4/Wnt reporter activity.

We apologize that we forgot to include this information on HCT116 cells. We have added this now at the beginning of the Results section, page 5: “HCT116 cells harbor a heterozygous stabilizing mutation in β -catenin, but these cells still respond to Wnt in reporter assays. Accordingly, we found that Wnt3a further increased and DKK1 treatment reduced basal TCF4/ β -catenin reporter activity. Co-treatment with the GSK3 β inhibitor CHIR99021 reverted this effect (Supplemental Fig. 1A).”

2. What I find amazing is that cells with stabilized beta-catenin are still dependent on Dkk, Lrp6 and GSK3. This must be due to Wnt/STOP of mitotic proteins, and has to be explained to the reader. Wnt/STOP is not even mentioned in the abstract or title.

We agree with the reviewer that downregulation of Wnt/STOP is responsible for the mitotic defects. We have re-written the abstract and now emphasize the importance of Wnt/STOP.

3. The remarkable finding is that basal GSK3 activity is regulated by Dkk/Lrp6, and as DKK decreases GSK3 sequestration in endosomes, the increase in GSK3 activity in the cytosol causes the cytogenetic abnormalities. That seems to be the main take home lesson of the paper: GSK3 activity is constantly downregulated by LRP6/Wnt-Porcupine signaling. The authors highlight the role of Wnt10b at the expense of the basal cellular activity of GSK3. This is a mistake in my view, because in these particular cells Wnt10b might be special but in others it will be other Wnts. Wnt10b is crucial in mammary gland and not that many other tissues, while GSK3 is universal. The title should mention GSK3 and Wnt/STOP in some way.

We have performed additional and more comprehensive experiments screening for Wnts that might be involved in the regulation of mitotic microtubule dynamics. This was a suggestion of reviewer #1. Here, we used Evi knockout cells, treated the cells with conditioned media for 14 different Wnts and investigated rescue activity for mitotic microtubule polymerization rates in living cells. We found that Wnt10b is the most potent Wnt for regulation of microtubule dynamics. However, we also found Wnt7a and Wnt9a as additional candidates with lower activity. This might indeed indicate, as the reviewer supposed, that Wnt10b is a relevant Wnt in

HCT116 cells and other Wnts (Wnt7a, Wnt9b) might be important in other cell types. We are presenting the new data now in our new Figure 4B and discuss the results in the Discussion section, page 11: *“In fact, we performed a comprehensive screen of 14 different human for their involvement in the regulation of microtubule dynamics in mitosis. We identified Wnt10b as a major Wnt ligand regulating mitotic microtubule dynamics, at least in HCT116 and in non-transformed RPE1 cells. In addition to that, also Wnt7b and Wnt9a might be candidate ligands with functions for mitosis, possibly in other cell types and/or cell lines.”*

We also agree with the reviewer in saying that, both, GSK3 and Wnt/STOP are highly relevant for the mitotic regulation. Therefore, we have changed the title accordingly: *“Wnt10b-GSK3 β -dependent Wnt/STOP signaling prevents aneuploidy in human somatic cells”*

4. An example of the assay of EB3 imaging to determine microtubule growth rates should be shown in the context of this paper in the EV supplemental information. The reader cannot be expected to refer back to the Stoltz et al. 2015, as is requested repeatedly. The quantitations are beautiful but some EV images are needed as well. Minor Comments:

We included a scheme explaining the principle of the microtubule growth rate measurements as Supplemental Figure 1D. In addition, we provide an example live cell movie as Supplemental Movie 1.

5. page 5 line 28, change demonstrating to suggesting. It is not demonstrated here.

We changed the wording.

6. page 9 lines 17-21 and Fig. 4C do not seem useful. Other cells will respond to other Wnts but in the end it is the increase in cytosolic GSK3 what counts.

We moved these data from Figure 4C to Supplemental Figure 3C and emphasize that the difference is seen in HCT116 cells (Results, page 9/10). In addition, we discuss the possibility that different Wnts can have different activity in different cells (Discussion, page 11, see above)

7. page 9, last paragraph, great experiment. How does one identify the mutant cells from the excess of parental cells?

We now provide a clear description in the Figure legend for Figure 4A: *“Mitotic microtubule plus end assembly rates in HCT116^{Evi_KO} cells after co-culture with parental HCT116 cells. HCT116^{Evi_KO} cells were transfected to express EB3-GFP and co-cultured with equal amount or 10-fold higher numbers of untransfected HCT116 cells for 48 hours. HCT116^{Evi_KO} cells were identified by EB3-GFP expression and microtubule polymerization rates were measured in mitotic cells.*

Scatter dot plots show average microtubule growth rates (20 microtubules/cell, mean +/- SEM, t-test, n=30 cells from three independent experiments)."

In addition this is also mentioned in the text (Results, page 8): *"To test whether proper mitotic regulation can be restored in HCT116^{Evi_KO} cells by paracrine Wnt signaling the knockout cells were co-cultured with parental HCT116 cells and microtubule growth rates were determined specifically in (EB3-GFP-transfected) HCT116^{Evi_KO} cells."*

8. In a paper just published, Albrecht et al. Cell Reports 2020, shows that Axin1 mutation or GSK3 inhibition triggers macropinocytosis and metabolite changes in a way **independent** of protein synthesis and beta-catenin stabilization. Axin1, APC and GSK3 repress macropinocytosis in basal cellular conditions. The basal levels of GSK3 activity seem to be very important in cell physiology.

We emphasize the importance of highly regulated and fine-tuned GSK3 activity and also included the mentioned paper by Albrecht et al. in our new Discussion section, page 11: *"It is interesting that fine-tuned regulation of Wnt associated GSK3 β activity seems not only be important for mitotic regulation, but also e.g. for macropinocytosis and metabolite changes as demonstrated recently [34]."*

9. The authors should discuss what would be the predicted effects of Dkk, Dkk antibodies or Porcupine inhibitors in a clinical setting as these seem to be promising treatments.

We have now included this important point into our new Discussion section, page 12/13: *"The link between Wnt signaling and aneuploidy might also have clinical implications. It is well established that hyper-active Wnt signaling has a tumor-promoting function [33]. Therefore, inhibition of Wnt signaling has been considered as an anti-cancer treatment strategy. In fact, several PORCN inhibitors, FZD antagonists, secreted frizzled-related proteins (SFRPs) or tankyrase inhibitors are currently tested in clinical phase1/2 trials for the treatment of various cancers [33]. Based on our results, these treatments would induce chromosomal instability and aneuploidy, conditions known to contribute to tumor progression by supporting tumor evolution and by facilitating the development of resistance towards chemotherapy [37]. This would put Wnt inhibitors at risk for the treatment of cancer and might even promote tumor progression. On the other hand, therapies aiming to activate Wnt signaling including DKK-neutralizing antibodies or GSK3 kinase inhibitors, have been considered for treatment of diseases with reduced Wnt signaling like Alzheimer's disease [36, 38]. Here, evolving aneuploidy, which is associated with such neurodegenerative diseases [39], might be indeed suppressed by Wnt activators as we have demonstrated in our cell-based studies."*

10. It is surprising that Davidson et al. Dev Cell 2009 is not even cited, for this study showed that Wnt signaling is elevated during the G2 and M phases of the cell cycle.

We have now included the Davidson et al reference in the introduction when

introducing cell cycle regulated Wnt signaling and Wnt/STOP (page 4).

11. Please say when first introduced that EV means Expanded View. I did not know.

The term “Expanded View” was used for the manuscript originally submitted to EMBO Reports. For publication in LSA, we are now using the term “Supplementary Figure”

November 15, 2020

RE: Life Science Alliance Manuscript #LSA-2020-00855-TR

Prof. Holger Bastians
University Medical Center Goettingen
GZMB Molecular Oncology
Grisebachstrasse 8
Göttingen 37077
Germany

Dear Dr. Bastians,

Thank you for submitting your revised manuscript entitled "Wnt10b-GSK3 β -dependent Wnt/STOP signaling prevents aneuploidy in human somatic cells". We would be happy to publish your paper in Life Science Alliance pending final revisions necessary to meet our formatting guidelines.

Along with the points listed below, please also attend to the following,

- please make sure the author order in your manuscript and in our system match
- please use the [10 author names, et al.] format in your references (i.e. limit the author names to the first 10)

A. FINAL FILES:

-- Summary blurb (enter in submission system): A short text summarizing in a single sentence the study (max. 200 characters including spaces). This text is used in conjunction with the titles of papers, hence should be informative and complementary to the title. It should describe the context and significance of the findings for a general readership; it should be written in the present tense

and refer to the work in the third person. Author names should not be mentioned.

B. MANUSCRIPT ORGANIZATION AND FORMATTING:

Sincerely,

Shachi Bhatt, Ph.D.
Executive Editor
Life Science Alliance
<https://www.lsjournal.org/>
Tweet @SciBhatt @LSAJournal

Reviewer #1 (Comments to the Authors (Required)):

The authors have addressed all critical points in an appropriate manner and added new data. The paper should now be acceptable for publication.

Minor point: There seems to be a mistake on p. 9, first sentence of new paragraph: "Wnt7a, Wnt9b"

should probably read "Wnt 7b, Wnt 9a".

Reviewer #2 (Comments to the Authors (Required)):

The paper has been greatly improved by the reviewing process, publish as is.

November 17, 2020

RE: Life Science Alliance Manuscript #LSA-2020-00855-TRR

Prof. Holger Bastians
University Medical Center Goettingen
Institute of Molecular Oncology
Grisebachstrasse 8
Goettingen 37077
Germany

Dear Dr. Bastians,

Thank you for submitting your Research Article entitled "Wnt10b-GSK3 β -dependent Wnt/STOP signaling prevents aneuploidy in human somatic cells". It is a pleasure to let you know that your manuscript is now accepted for publication in Life Science Alliance. Congratulations on this interesting work.

DISTRIBUTION OF MATERIALS:

Again, congratulations on a very nice paper. I hope you found the review process to be constructive and are pleased with how the manuscript was handled editorially. We look forward to future exciting submissions from your lab.

Sincerely,

Shachi Bhatt, Ph.D.

Executive Editor

Life Science Alliance

<https://www.lsjournal.org/>
